# Lumbar Spinal Stenosis Treatment: Is Surgery Better than Non-Surgical Treatments in Afro-Descendant Populations?

**DOI:** 10.3390/biomedicines10123144

**Published:** 2022-12-06

**Authors:** Fabienne Louis-Sidney, Jean-Florent Duby, Aïssatou Signate, Serge Arfi, Michel De Bandt, Benoit Suzon, Philippe Cabre

**Affiliations:** 1Service de Rhumatologie, CHU de Fort de France, 97200 Fort de France, France; 2Service de Neurologie, CHU de Fort de France, 97200 Fort de France, France; 3Service de Médecine Interne, CHU de Fort de France, 97200 Fort de France, France

**Keywords:** lumbar spine stenosis, surgery, conservative treatments, afro-descendants

## Abstract

(1) Background: Limited data are available on lumbar spine stenosis management in sub-Saharan African populations and Afro-descendant patients are underrepresented in European and US clinical trials. We aimed to compare the clinical response between decompressive surgery and conservative treatments in a population of self-reported Afro-Caribbean patients with lumbar spine stenosis over a 2-year follow-up period. (2) Methods: Prospective cohort of 137 self-reported Afro Caribbeans with lumbar spine stenosis based on clinical and radiological criteria. Patients were assigned to decompression surgery or to conservative treatments according to their outcome after a first course of steroid epidural injection and their preferences. The primary outcome was evolution of the Oswestry disability index at 3 months (3 M), 12 M, 18 M and 24 M follow-up. (3) Results: Decrease of ODI was significantly more important in the “decompression surgery” arm compared to “conservative treatment” arm at 3 M, 12 M and 18 M: −17.36 vs. 1.03 *p* < 10^−4^; −16.38 vs. −1.53 *p* = 0.0059 and −19.00 vs. −4.52 *p* = 0.021, respectively. No difference was reported at 24 M. (4) Conclusions: In this first comparative study between surgery and conservative treatments in an exclusively afro-descendant lumbar spine stenosis cohort, we report long term superiority of decompression surgery versus conservative treatments over an 18-month period.

## 1. Introduction

Lumbar spine stenosis is one of the most frequent conditions requiring spine surgery [1] and therefore represents a substantial cost to public health. Superiority of surgical treatment over conservative approaches has been reported in several studies with a large number of surgical and conservative procedures and numerous criteria used to evaluate treatment efficiency were used for evaluation. [2,3,4]. However, a meta-analysis from the Cochrane Library in 2016, suggested that current evidence comparing surgery to non-surgical approaches in lumbar spine stenosis was of low quality and that standardized conservative protocols were needed [5].

Moreover, limited data are available on lumbar spine stenosis management in sub-Saharan African populations, despite this being one of the most common rheumatological conditions in the region [6] and despite an increased frequency of congenital narrow lumbar canals in this population [7]. Only one retrospective study demonstrated efficiency of surgery in a Cameroonian population with lumbar spine stenosis [8] and Afro-descendant patients remain heavily underrepresented in European and US clinical trials [4,9,10]. In the retrospective lumbar spine stenosis cohort of Elsamadicy et al., the African American subgroup was less likely to be treated surgically than the white subgroup, despite presenting with more severe symptoms [11]: higher Oswestry disability index (ODI), higher leg pain visual analogic scale (VAS-LP) and back pain visual analogic scale (VAS-BP). Response to surgery was comparable between both subgroups on the evaluation criteria (ODI VAS-LP, VAS-BP) but satisfaction was significantly lower in the Afro-American subgroup. Jancuska et al. reported a lower proportion of African American patients in New York State undergoing surgery for lumbar spine stenosis compared to white populations [12]. There are numerous underlying factors behind this under-representation and disparities in care access, but the socio-economic context affecting a substantial part of African descendants living in the American and European continents should not be ignored.

Martinique is a French Caribbean region with a population of mostly sub-Saharan African origin and a healthcare system comparable to mainland France. Its social system provides free access to care for all patients regardless of their income.

TELEMAR is a prospective cohort of Martinican patients with lumbar spine stenosis, recruited between 2001 and 2005, with a 2-year clinical follow-up [13]. A previous study reported a favorable response in 25.9% of patients at 3 months of two steroid epidural injections course using ODI as the primary outcomes [13]. Among these patients, we aimed to compare the clinical response between decompressive surgery and conservative treatments in a population of self-reported Afro-Caribbean patients with lumbar spine stenosis over a 2-year follow-up period after the first evaluation.

## 2. Materials and Methods

### 2.1. Population and Study Design

TELEMAR is a prospective cohort of Martinican patients presenting with lumbar spine stenosis based on clinical and radiological criteria as previously described [12]. Among 138 patients evaluated in phase I of the study and eligible for inclusion in phase II of the study, 137 self-reported being Afro-Caribbeans.

Phase I consisted in a clinical evaluation of patients based on ODI, VAS-LP and VAS-BP, 3 months after a first course of two epidural steroids injection and standardized physiotherapy. Patients were assigned to three different arms, according to their outcome and willing preferences at the 3 months follow-up visit of phase I. 

Exclusion criteria were:--Having already undergone spinal surgery or nucleolysis--Presenting with stroke sequels or any other neurological condition causing a severe motor handicap--Presenting with cognitive difficulties that make the evaluation uncertain--Presenting other musculoskeletal conditions causing severe disability

A clinical follow-up with standardized physiotherapy prescription was proposed to patients who were considered improved at phase I (decrease in ODI greater than 20%). A second course of epidural steroid injections was recommended to patients with stable ODI (−20% to 20% changes in ODI) and surgery was proposed to those with poor outcomes (ODI increase of more than 20%) at the end of Phase I. Patients with clinical follow-up and a second course of epidural steroid injection were pooled into a single “conservative treatments” arm. Each patient had a 3 M, 12 M, 18 M and 24 M follow-up visit. During each visit, patients had to complete the ODI questionnaire and indicate leg pain and back pain using a visual analogic scale. ODI is an outcome measure that assesses function in activities of daily living for patients with back pain and has been used in numerous studies evaluating clinical evolution in lumbar spine stenosis [14,15,16].

Patients in the conservative treatment who then underwent surgery during follow-up were considered as withdrawals.

Patients were also asked for self-reported improvement (binary response: yes or no).

Investigations were carried out following the rules of the Declaration of Helsinki of 1975, revised in 2013, and our study received approval from the institutional review board (IRB) of the University Hospital of Fort de France.

### 2.2. Outcomes

The primary outcome was change in ODI (ΔODI) at 3 months (M3), M12, M18 and M24 of conservative treatments or surgery. Change in ODI was defined by the difference between ODI at time of evaluation and ODI at baseline. Secondary outcomes were changes in absolute value of VAS-BP (ΔVAS-BP) with scoring range from 0 to 100 mm and VAS-LP (ΔVAS-LP) with scoring range from 0 to 100 mm, and a binary self-reported improvement at M3, M12, M18 and M24.

### 2.3. Statistics

Quantitative variables were described as mean with standard deviation. Qualitative variables were described as number of patients and percentage. Univariate comparative analysis was performed with Fisher’s test for percentages and Student’s test for means. Multivariate analyses were performed using logistic regression and results were presented using odds ratios with 95% confidence intervals. The statistical relationship between ODI and VAS-LP at baseline was reported with a Pearson’s correlation coefficient r = 0.48 *p* < 10^−4^. ΔODI was adjusted on baseline ODI and ΔVAS-LP was adjusted on VAS-LP at baseline.

## 3. Results

A flow chart of the study population is presented in Figure 1.

### 3.1. Population

On 137 patients eligible for second phase of study, 132 were included.

Surgical decompression, second course of epidural steroid injections and clinical follow-up were proposed to 27, 69 and 36 patients, respectively.

After patients’ wishes were considered, 25, 36, and 71 patients were offered surgery, epidural steroids injections and clinical follow-up, respectively. Patients from the clinical follow-up and epidural steroids injections were pooled into a single “conservative treatments” arm.

Mean age of population was 62.5 (±13.2) years. A total of 55 (41.67%) patients were male. Concerning occupational status, 79 (14.4%) patients were retired, 14 (10.6%) unemployed, 18 (13.6%) on medical leave and 21 (15.9%) were full or part-time workers.

Body mass index (BMI) was above 25 kg/m^2^ in 70.63% of cases. Sixty-seven (50.76%) and twenty-nine (21.97%) patients had high blood pressure (HBP) and diabetes mellitus (DM) respectively. Multilevel lumbar stenosis was reported in 48 (36.4%) patients of the cohort. Three cohort patients had lumbar spine stenosis with an associated lumbar spondylolisthesis.

A comparative analysis of “conservative treatments” arm and “surgery” arm is presented in Table 1. At baseline, both arms were comparable for age, male/female ratio, BMI, HBP, DM and duration of symptoms (*p* = 0.86, *p* = 0.38, *p* = 0.72, *p* = 1, *p* = 0.12, *p* = 0.41 respectively). The Owestry disability index and VAS-LP were significantly higher and foraminal stenosis significantly more frequent in the “decompression surgery” arm (*p* = 0.02, *p* = 0.0006, *p* = 10^−3^, respectively).

No significant difference between both arms was reported for VAS-BP (*p* = 0.11).

### 3.2. Changes in ODI (ΔODI), VAS-LP (ΔVAS-LP) and VAS-BP (ΔVAS-BP)

Comparative ΔODI, ΔVAS-LP and ΔVAS-BP between “conservative treatments” and “surgery” arms are presented in Table 2.

Decrease of ODI was significantly more important in the “decompression surgery” arm compared to “conservative treatment” arm at 3 M, 12 M and 18 M: *p* < 10^−4^, *p* = 0.0059 and *p* = 0.021, respectively.

Decrease in VAS-LP was significantly higher in the “surgery” arm compared to “conservative treatment” arm at 3 M and 18 M: *p* < 10^−4^ and *p* = 0.0012, respectively. A tendency to higher VAS-LP decreases in the “surgery” arm at 12 M (*p* = 0.09).

Self-reported improvement was correlated to decrease of ODI and VAS-LP, with a significant higher proportion of patients reporting it in the “surgery” arm at 3 M and 18 M (*p* = 0.0036 and *p* = 0.048) and a tendency at 12 M (*p* = 0.06).

No difference in VAS-BP evolution was observed at any time of follow-up between both arms.

Multivariate analyses of ODI and VAS-LP changes are presented in Table 3.

Adjusted on baseline ODI, significant higher decrease was observed for ODI at 3 M, 12 M and 18 M in the “surgery” arm (*p* = 0.048 at 3 M, 12 M and 18 M).

Adjusted on baseline VAS-LP, significant higher decrease was observed for VAS-LP at 3 M and 18 M (*p* = 10^−4^ and *p* = 0.0068, respectively), and a tendency for higher decrease at 12 M (*p* = 0.07). No significant difference was observed for ΔODI or ΔVAS-LP at 24 M.

In the surgery arm, ODI at 3 M, 12 M, 18 M and 24 M was 30.52 (±18.09), 32.33 (±18.22), 28.56 (±16.76) and 37 (±19.06), respectively.

In that same arm, VAS-LP at 3 M, 12 M, 18 M and 24 M was 26.76 (±23.63), 35.95 (±29.53), 30.05 (±17.82) and 40.88 (±32.27), respectively; and VAS-BP was 34.72 (±26.89), 31.38 (±20.39), 36.18 (±24.15) and 41.24(±25.09), respectively.

No difference was reported between surgery and conservative treatment arms for ODI, VAS-LP and VAS-BP, except for VAS-LP at 3 M, which was significantly higher in the conservative treatments arm compared to the surgery arm (*p* = 0.01).

### 3.3. Surgical Techniques and Complications

Seventeen surgical reports were available: all 17 patients underwent decompression laminectomy surgery with associated fusion for 6 of them.

Three patients presented early complications: a torn redon drain requiring immediate revision surgery, a dural breach and an abnormality on suture repaired two weeks later.

## 4. Discussion

To our knowledge, this is the first comparative study between surgery and conservative treatments in an exclusively Afro-descendant lumbar spine stenosis cohort. 

Surgery was superior to conservative treatment in our lumbar spine stenosis cohort as described in previous European, US and Asian comparative studies [4,10,17,18]. 

We report long term improvements of ODI, leg pain, and long-term satisfaction of patients undergoing surgery with more than 90% of the surgery population self-reporting improvement and presenting a sustainable decrease of ODI and leg pain VAS. 

This superiority of surgery above conservative treatments, including a second course of epidural steroids injection, raises questions about pertinence of repeated steroid epidural injections in lumbar spine stenosis that are commonly used for patients presenting with the condition. Moreover, studies, reviews and meta-analyses [19,20,21,22] have already reported epidural steroid injections to provide moderate and short term improvement of patients with lumbar spine stenosis. 

Contrary to some other studies [4,16], no improvement of back pain was observed in our cohort at any time. ODI is a functional score evaluating impact of back pain on daily life activities. Discordance between ODI and back pain highlights importance of functional outcomes measures in evaluation of patients with lumbar spine stenosis. 

ODI is one of the most used functional outcomes in lumbar spine stenosis studies. Other specific outcomes measures such as the Roland-Morris disability questionnaire, Swiss spinal stenosis questionnaire, Self-Paced Walk Test and less specific SF-36 outcomes measure have been used in numerous studies [23] after 2008. At the time of constituting our cohort, ODI was the most used functional outcomes measure, explaining the use of this single functional outcome measure. 

Concerning the moderate proportion of minor complications in our surgery cohort (17% of patients presenting with early post-surgery complications), the small size sample of patients might be an explanation for this phenomenon. 

An interesting point is comparison of our cohort with African and African American patients undergoing the same condition. In Bello et al., for an African surgery cohort and Elsamadicy et al., for an African American surgery cohort, the age at surgery was respectively 58.2 and 54 years [8,11], whereas in our surgery, the cohort age was 62.9 years. Younger age at diagnosis of lumbar spine stenosis might reflect the potential role of congenital narrowing of the lumbar spine. The age in our surgery cohort tended to be closer to those of the European and US cohorts: 63.8 years in the cohort of Weinstein et al. [4] and 63 years in the cohort of Slätis et al. [10].Another point was the presence of lower ODI in our surgery cohort (47.9%) and in the African American cohort (25%) [11] compared to the African cohort (78%) [8]. Such clinical heterogeneities highlight a potential role of environment, including quality of life and psychosocial factors, as determinants for phenotypical expression of the disease.

Our study presents with some limitations. First, the high proportion of lost to follow-up (47.2% in the conservative treatments arm and 42.9% in the surgery arm at 24 M) might have decreased the study power explaining absence of significance and some discordance in results at 24 M. Surgical management was at the surgeons’ discretion and not all patients were operated on in Martinique. Therefore, there was not homogeneity in the surgical techniques.

Our study has some strengths also. In this observational cohort, in a real-life approach of a complex problem, treatment assignment of patients was based on each patient’s severity and preferences. This is consistent with what is observed in real life, and this was an ethical choice made by our research team. Physiotherapy prescriptions in both clinical follow-up and second epidural steroids injection groups were standardized for every patient. Finally, we provide long terms results in a small but consistent cohort of lumbar spine stenosis.

## 5. Conclusions

In this first comparative study between surgery and conservative treatments in an exclusively Afro-descendant lumbar spine stenosis cohort, we report long term superiority of surgery versus conservative treatments over an 18-month period.

## Figures and Tables

**Figure 1 biomedicines-10-03144-f001:**
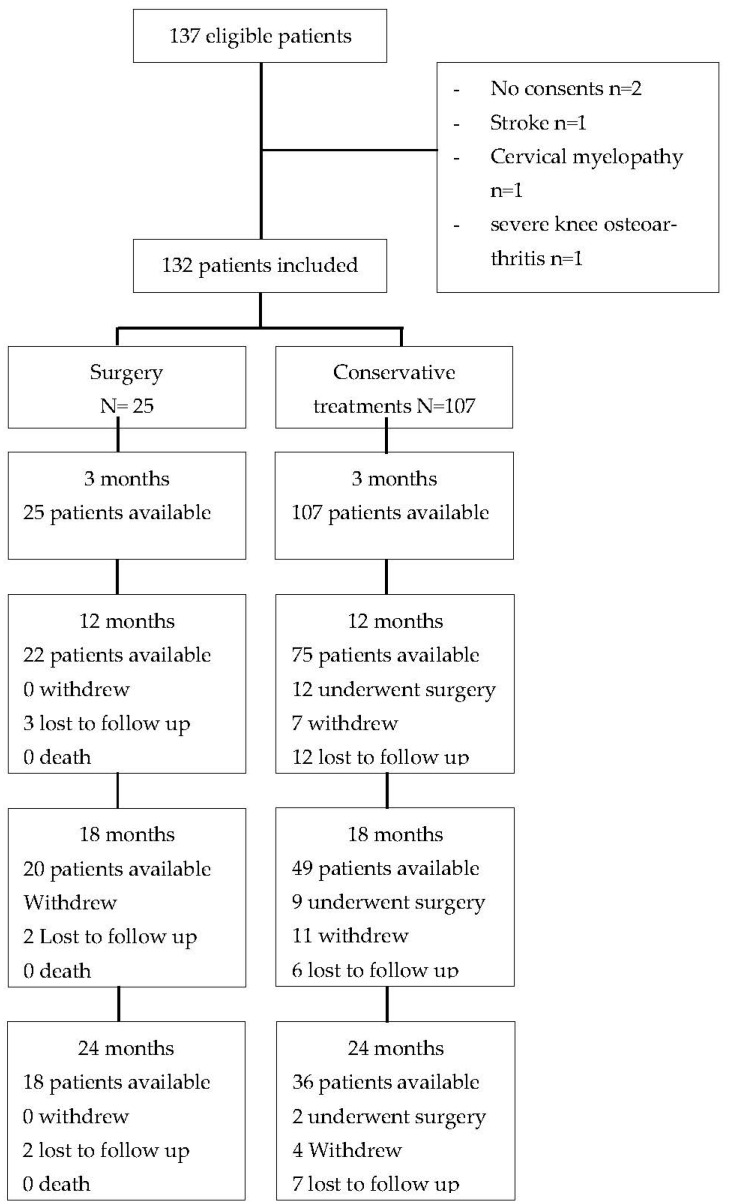
Flow chart of study population.

**Table 1 biomedicines-10-03144-t001:** Population characteristics at baseline.

Characteristics	Conservative Treatments (N = 102)	Surgery (N = 25)	*p*
Age—mean (SD)	62.4 (±13.5)	62.9 (±12.2)	0.86
Male—N (%)	47 (43.9)	8 (32)	0.38
BMI—mean (SD)	27.7 (±4.25)	27.3 (4.9)	0.72
HBP—N (%)	58 (54.2)	9 (36)	1.00
Diabetes—N (%)	24 (22.4)	5 (20)	0.12
ODI—mean (SD)	37.9 (±17.9)	47.9 (±21.3)	**0.02**
VAS-BP—mean (SD)	39.8 (±23.4)	49 (±29.35)	0.11
VAS-LP—mean (SD)	45.7 (±26.3)	66.7 (24.6)	**0.0006**
Duration of symptoms (years)—mean (SD)	4.2 (±5.5)	3.3 (±3.1)	0.41
Central stenosis	97 (90.6)	23 (92)	1.00
Lateral recess stenosis	86 (80.4)	20 (80)	1.00
Foraminal stenosis	24 (22.4)	16 (64)	**10^−3^**
2nd epidural steroids injection—N (%)	36 (27.3)	-	-
Clinical follow-up—N (%)	71 (53.8)	-	-

BMI: body mass index, HBP: High blood pressure, ODI: Oswestry disability index, VAS-LP: Visual analogic scale leg pain, VAS-BP: Visual analogic scale back pain. Bolded results match with significant findings.

**Table 2 biomedicines-10-03144-t002:** Changes in Oswestry disease index, VAS back pain and VAS leg pain at 3,12,18 and 24 months (univariate analysis).

		Conservative Treatments N = 107	Surgery N = 25	*p*
3 months	ΔODI—mean (SD)	1.03 (13.48)	−17.36 (21.91)	**<10^−4^**
	ΔVAS-BP—mean (SD)	−1.78 (24.50)	−10.36 (30.95))	0.21
	ΔVAS-LP—mean (SD)	−6.00 (25.83)	−43.30 (28.11)	**<10^−4^**
	Self-reported improvement—N (%)	63 (61.17)	23 (92)	**0.0036**
12 months	ΔODI—mean (SD)	−1.53 (15.75)	−16.38 (21.02)	**0.0059**
	ΔVAS-BP—mean (SD)	0.25 (25.9)	−8.29 (34.13)	0.30
	ΔVAS-LP—mean (SD)	−6.05 (27.31)	−22.95 (41.44)	0.09
	Self-reported improvement—N (%)	36 (54.55)	16 (80)	0.06
18 months	ΔODI—mean (SD)	−4.52 (14.99)	−19.00 (18.59)	**0.021**
	ΔVAS-BP—mean (SD)	−3.75 (33.26)	−2.41 (27.45)	0.88
	ΔVAS-LP—mean (SD)	−2.20 (26.25)	−29.47 (32.36)	**0.0012**
	Self-reported improvement—N (%)	29 (69.05)	16 (94.12)	**0.048**
24 months	ΔODI—mean (SD)	−6.85 (13.34)	−13.22 (20.87)	0.25
	ΔVAS-BP—mean (SD)	−5.22 (24.25)	−1.94 (39.02)	0.72
	ΔVAS-LP—mean (SD)	−13.97 (25.29)	−21.35 (34.78)	0.39
	Self-reported improvement—N (%)	26 (76.47)	10 (62.50)	0.33

ΔODI: changes in Oswestry disease index, ΔVAS-BP: changes in VAS back pain, ΔVAS-LP: changes in VAS leg pain. Bolded results match with significant findings.

**Table 3 biomedicines-10-03144-t003:** Evolution of Oswestry disease index and leg pain VAS at 3, 12, 18 and 24 months (multivariate analysis).

	Conservative Treatments N = 107	Surgery N = 25	OR	IC_95_	*p*
3 M ΔODI—mean (SD)	1.03 (13.48)	−17.36 (21.91)	0.95	0.916–0.980	**0.048**
12 M ΔODI—mean (SD)	−1.53 (15.75)	−16.38 (21.02)	0.965	0.933–0.998	**0.048**
18 M ΔODI—mean (SD)	−4.52 (14.99)	−19.00 (18.59)	0.94	0.897–0.998	**0.048**
24 M ΔODI—mean (SD)	−6.85 (13.34)	−13.22 (20.87)	1.00	0.960–1.042	0.99
3 M ΔVAS-LP—mean (SD)	−6.00 (25.83)	−43.30 (28.11)	0.945	0.918–0.973	**10^−4^**
12 M ΔVAS-LP—mean (SD)	−6.05 (27.31)	−22.95 (41.44)	0.976	0.951–1.002	0.07
18 M ΔVAS-LP—mean (SD)	−2.20 (26.25)	−29.47 (32.36)	0.944	0.906–0.984	**0.0068**
24 M ΔVAS-LP—mean (SD)	−13.97 (25.29)	−21.35 (34.78)	0.994	0.969–1.020	0.66

ΔODI: changes in Oswestry disease index, ΔVAS-LP: changes in VAS leg pain, 3 M: at 3 months, 12 M: at 12 months, 18 M: at 18 months, 24 M: at 24 months. Bolded results match with significant findings.

## Data Availability

Data sharing is available.

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
