# Peer review of "Lumbar Spinal Stenosis Treatment: Is Surgery Better than Non-Surgical Treatments in Afro-Descendant Populations?"

_biomedicines, 2022, doi:10.3390/biomedicines10123144_

Round 1

Reviewer 1 Report

General impression

In this article, the authors aimed to compare the clinical response between decompressive surgery and conservative treatments in a population of self-reported Afro-Caribbean patients with lumbar spine stenosis over a 2-year follow-up period after the first evaluation.  And they concluded that long term superiority of decompression surgery versus conservative treatments over an 18-month period.  As the authors mentioned, this report must be valuable at the point that such kind of research targeted sub-Saharan African population has not been done so much ever.

The methodology of this study was precisely explained.  Also, the limitations of this project were indicated.  I could not find either error in writings or mistakes in the text.  However, I have some minor requests to be revised as stated below.  After they have been resolved, I will judge this manuscript can be accepted and published by biomedicines journal.

1. Results  page 3 line 110

I think “Flow Charts” must be used in plural.

I guess “Flow Charts is presented in Figure 1.” may be changed to “The Flow Chart is presented in Figure 1.”.

2. Figure 1. Flow Chart of study population

The last word of explanation about excluded patients in the square is broken in the middle.

“severe knee osteoar-” should be replaced by “severe knee osteoarthritis”.

3. Results  page 5 line 141-145

I think these sentences are repeated.  They may be deleted.

4. Results  page 7 line 176-177

A space should be inserted between line 176 and 177.

5. Discussion  page 8 line 205-205, 218-220

  Sentences at page 8 line 205-205 and line 218-220 are almost same.  Either one can be deleted.

6. Discussion  page 8 line 231

  The sentence here; “Our study presents with some limitations.” should be begun on new line.

Author Response

Author 2

General impression

In this article, the authors aimed to compare the clinical response between decompressive surgery and conservative treatments in a population of self-reported Afro-Caribbean patients with lumbar spine stenosis over a 2-year follow-up period after the first evaluation.  And they concluded that long term superiority of decompression surgery versus conservative treatments over an 18-month period.  As the authors mentioned, this report must be valuable at the point that such kind of research targeted sub-Saharan African population has not been done so much ever.

The methodology of this study was precisely explained.  Also, the limitations of this project were indicated.  I could not find either error in writings or mistakes in the text.  However, I have some minor requests to be revised as stated below.  After they have been resolved, I will judge this manuscript can be accepted and published by biomedicines journal.

  1. Results  page 3 line 110

I think “Flow Charts” must be used in plural.

I guess “Flow Charts is presented in Figure 1.” may be changed to “The Flow Chart is presented in Figure 1.”.

 Corrections were made as requested.

  1. Figure 1. Flow Chart of study population

The last word of explanation about excluded patients in the square is broken in the middle.

“severe knee osteoar-” should be replaced by “severe knee osteoarthritis”.

 Corrections were made as requested.

  1. Results  page 5 line 141-145

I think these sentences are repeated.  They may be deleted.

 Corrections were made as requested.

  1. Results  page 7 line 176-177

A space should be inserted between line 176 and 177.

 Corrections were made as requested.

  1. Discussion  page 8 line 205-205, 218-220

  Sentences at page 8 line 205-205 and line 218-220 are almost same.  Either one can be deleted.

 Corrections were made as requested.

  1. Discussion  page 8 line 231

  The sentence here; “Our study presents with some limitations.” should be begun on new line.

Corrections were made as requested.

Reviewer 2 Report

Introduction.

They wrote: "To date no clear superiority of surgical treatment over conservative approaches has been reported: meta-analyses do not provide any consensus...". This sentence is suported by the reference "Zaina, F.; Tomkins-Lane, C.; Carragee, E.; Negrini, S. Surgical versus Non-Surgical Treatment for Lumbar Spinal Stenosis. 267 Cochrane Database Syst. Rev. 2016, 2016, CD010264, doi:10.1002/14651858.CD010264.pub2." which, for its part, has received serious criticisms because "the quality of the evidence for all outcomes was graded low due to high risk of bias, study design, and imprecision due to incomplete outcome data. According to the GRADE approach, low-quality evidence indicates little confidence in the effect estimate and that the true effect is likely to be substantially different from the estimate of effect" (Aleem IS, Drew B. Cochrane in CORR ®: Surgical Versus Non-surgical Treatment for Lumbar Spinal Stenosis. Clin Orthop Relat Res. 2017 Nov;475(11):2632-2637. doi: 10.1007/s11999-017-5452-0. Epub 2017 Jul 28. PMID: 28755156; PMCID: PMC5638738.). Moreover, "Now with 8-year results, the Spine Patient Outcomes Research Trial (SPORT) represents the largest and highest-quality study available". This article is included in the bibliography of the paper under evaluation. I suggest the authors moderate the discussion of the lack of evidence for the superiority of surgical treatment and include Weinstein's paper in the introduction.

They argue as a unique contribution that all their patients are of African origin. But they do not explain why they expect that this population group might have different outcomes from the treatment of canal stenosis than other ethnic groups.

Results. Considering the detail with which the results are presented in Tables 1 and 2, the length of the text could be reduced as it is redundant with respect to the tables.

Table 3 should be modified. The legend includes an explanation of the logistic regression adjustment method. This should be included in the statistical analysis section.

Discussion.

Page 8 Line 213 Roland-Morris instead of Rolland Morris

Author Response

They wrote: "To date no clear superiority of surgical treatment over conservative approaches has been reported: meta-analyses do not provide any consensus...". This sentence is suported by the reference "Zaina, F.; Tomkins-Lane, C.; Carragee, E.; Negrini, S. Surgical versus Non-Surgical Treatment for Lumbar Spinal Stenosis. 267 Cochrane Database Syst. Rev. 2016, 2016, CD010264, doi:10.1002/14651858.CD010264.pub2." which, for its part, has received serious criticisms because "the quality of the evidence for all outcomes was graded low due to high risk of bias, study design, and imprecision due to incomplete outcome data. According to the GRADE approach, low-quality evidence indicates little confidence in the effect estimate and that the true effect is likely to be substantially different from the estimate of effect" (Aleem IS, Drew B. Cochrane in CORR ®: Surgical Versus Non-surgical Treatment for Lumbar Spinal Stenosis. Clin Orthop Relat Res. 2017 Nov;475(11):2632-2637. doi: 10.1007/s11999-017-5452-0. Epub 2017 Jul 28. PMID: 28755156; PMCID: PMC5638738.). Moreover, "Now with 8-year results, the Spine Patient Outcomes Research Trial (SPORT) represents the largest and highest-quality study available". This article is included in the bibliography of the paper under evaluation. I suggest the authors moderate the discussion of the lack of evidence for the superiority of surgical treatment and include Weinstein's paper in the introduction.

Corrections were made as requested.

They argue as a unique contribution that all their patients are of African origin. But they do not explain why they expect that this population group might have different outcomes from the treatment of canal stenosis than other ethnic groups.

Response: Thank you for your very interesting remark.

In the introduction, we pointed out that after surgery, satisfaction was lower in the African American sub-group compared to the caucasian sub-group despite ODI being similar in both subgroups.
As being an afro-descendant population, we could expect similar results in the Martinican population, but we emphasize the potential role of socio-economic determinants in the results of the US study and that race might not be the major factor explaining differences. Race would presuppose a biological or even genetic substrate, ethnicity would refer to cultural markers, i.e. to an origin defined by the cultural community. Martinican and African American shares racial (and genetic) and some cultural backgrounds but part of the history and socio economics factors of those populations are different.
Contrary to African American, we can see in our study a good correlation between decrease of ODI and global satisfaction. This difference suggests a potential role of socio-economic determinants.   

 Results. Considering the detail with which the results are presented in Tables 1 and 2, the length of the text could be reduced as it is redundant with respect to the tables.

One sentence was repeated, so we deleted it.
Only p  were conserved in the text describing the results in the tables.

Table 3 should be modified. The legend includes an explanation of the logistic regression adjustment method. This should be included in the statistical analysis section.

 Corrections were made as requested.

Discussion.

Page 8 Line 213 Roland-Morris instead of Rolland Morris

Corrections were made as requested.